# Coastal-Cosmo-Model (CCMv1): a cosmogenic nuclide model for rocky coastlines

Richard S. Jones<sup>1</sup>

<sup>1</sup>School of Earth Atmosphere and Environment, Monash University, Clayton, 3800, Victoria, Australia

Correspondence to: Richard S. Jones (richard.s.jones@monash.edu)

Abstract. Understanding the long-term evolution of rocky coasts requires models that can account for complex interactions between exposure, erosion and sea level, constrained by empirical observations. This paper introduces Coastal-Cosmo-Model version 1 (CCMv1), a modular forward modelling framework designed to reconstruct coastal histories from in situ cosmogenic nuclide concentrations. CCMv1 integrates community-standard production rate calculations and allows flexible inversion of platform histories using discrete erosion and exposure parameters. The model includes four sub-models—inheritance, zero erosion, down-wearing only, and cliff retreat with down-wearing—enabling users to test hypotheses of increasing complexity. Crucially, CCMv1 can be applied to both eroding and non-eroding coastlines, offering a means to investigate the dominant controls on rocky shore histories for different settings. A demonstration using a published dataset from shore platform shows that CCMv1 effectively reproduces measured nuclide concentrations and supports a history of Holocene cliff retreat. CCMv1 provides an adaptable and hypothesis-driven framework for exploring rocky shore histories, with potential for future development to incorporate probabilistic optimisation and additional nuclide systems, and implementation for testing complex (multi-stage) erosion histories or relative sea-level histories.

# 1 Introduction

Cosmogenic nuclide analysis is a powerful method for reconstructing the erosional history of rocky coasts (Trenhaile, 2018). This is because platform formation by cliff retreat and down-wearing leaves a measurable imprint on surface nuclide concentrations (e.g. Choi et al., 2012). However, interpreting these concentrations requires numerical modelling to account for the complex interplay between nuclide production, erosion, sea-level change, tidal regime and beach cover. Existing forward models have explored how individual processes influence nuclide concentrations (e.g. Hurst et al., 2017; Regard et al., 2012), and have been applied to infer past coastal erosion rates by fitting model predictions to observations—determining how platforms evolved and testing if modern erosion rates have surpassed that of past millennia (e.g. Hurst et al., 2016; Swirad et

al., 2020).

While these models provide valuable insights, they often rely on fixed process formulations, can be restrictive in the mechanisms that can be explored, may not fully capture time-dependent shielding effects at depth, and are typically not coupled

35

to a widely-used production rate calculation framework. To address these limitations and provide an alternative approach, I present a forward modelling framework with empirical optimisation for predicting *in situ*-produced cosmogenic nuclide (e.g. <sup>10</sup>Be) concentrations across shore platforms through time. This paper describes CCMv1, a model that:

- 1. Allows flexible parameter fitting by optimising discrete erosion and exposure parameters, rather than enforcing predefined process relationships.
- 2. Uses community-standard nuclide production calculations via CRONUScalc (Marrero et al., 2016) and the global calibration dataset (Borchers et al., 2016), enabling accurate calculation of nuclide concentrations from time- and depth-dependent spallation and muon production.
- 3. Integrates forwarding modelling within an optimisation framework to effectively invert for the exposure, erosion and shielding history that best fits measured nuclide concentrations.

This approach enables an adaptable reconstruction of platform evolution as a function of cliff retreat, down-wearing and sea-40 level change, incorporating spatially and temporally varying topographic, water and cover shielding effects.

## 2 Model framework

CCM consists of four models to be used for different purposes, requiring different inputs and assessing different variables (Table 1). A cosmogenic inheritance model finds the best-fit surface erosion rate for a measured inheritance sample assuming steady-state production (Hurst et al., 2016), which can be used to evaluate whether the sample represents a feasible inheritance value for the site; if a best-fit erosion rate is found, the inheritance can be justifiably used to correct platform nuclide concentrations prior to modelling the platform erosion history.

A series of platform erosion models are designed to then test hypotheses of increasing complexity. The simplest hypothesis is that there is no relationship between erosion and the nuclide concentrations (i.e. the platform is a stable feature and no erosion has occurred), whereas the most complex hypothesis is that the nuclide concentrations are the result of cliff retreat, down-wearing and other factors over time. The simplest hypotheses should be ruled out before advancing to a more complex hypothesis to explain the data. A zero erosion model, which includes no cliff retreat or down-wearing, finds the best-fit total exposure time from a given sea-level history, with or without surface cover (e.g. beach). A down-wearing model, with no cliff retreat, finds the best-fit total exposure time from a given sea-level history and present-day down-wearing rate, with or without surface cover, keeping the present-day rate constant through time or finding a best-fit past rate relative to present (accelerating, decelerating or constant). Finally, a cliff retreat model (including down-wearing) finds the best-fit total exposure time and cliff retreat rate (accelerating, decelerating or constant), from a given sea-level history, present-day cliff retreat and down-wearing rates, with or without surface cover.

 Table 1. Suite of CCM models to evaluate a cosmogenic nuclide dataset.

| Model                    | Required inputs                                                                                                                                                                                                                                                                   | Free parameters (require initial value)                                                                                    | Outputs (best-fit result and variables)                                                                                                                                                                                                                                                                                                                           |
|--------------------------|-----------------------------------------------------------------------------------------------------------------------------------------------------------------------------------------------------------------------------------------------------------------------------------|----------------------------------------------------------------------------------------------------------------------------|-------------------------------------------------------------------------------------------------------------------------------------------------------------------------------------------------------------------------------------------------------------------------------------------------------------------------------------------------------------------|
| Cosmogenic inheritance   | Cosmogenic nuclide samples; Depth of inheritance sample from cliff top                                                                                                                                                                                                            | Cliff surface erosion rate                                                                                                 | Surface erosion rate; nuclide production profile (from spallation, muons and total)                                                                                                                                                                                                                                                                               |
| Zero platform<br>erosion | Cosmogenic nuclide samples; Topographic shielding across the platform; Elevation profile of the platform; Relative sea- level history; Tidal data and benchmarks; Measurements to use for misfit calculation;  Density of surface cover (optional)                                | Total model time;  Surface cover depth  (optional)                                                                         | Total exposure (model time); nuclide concentrations across platform; platform submergence time; cumulative water shielding and topographic shielding; optionally, mean surface cover depth and cumulative cover shielding                                                                                                                                         |
| Down-wearing only        | Cosmogenic nuclide samples; Topographic shielding across the platform; Elevation profile of the platform; Relative sea- level history; Tidal data and benchmarks; Present-day down-wearing rate; Measurements to use for misfit calculation;  Density of surface cover (optional) | Total model time;  Past rate of down- wearing (multiplier relative to present) (optional);  Surface cover depth (optional) | Total exposure (model time); platform profile prior to down- wearing; nuclide concentrations across platform; platform submergence time; cumulative rate of down-wearing across profile; cumulative water shielding and topographic shielding; optionally, past rate of down-wearing relative to present, mean surface cover depth and cumulative cover shielding |

| Cliff retreat and down-wearing | Cosmogenic nuclide samples; Topographic shielding across the platform; Elevation profile of the platform; Relative sea- level history; Tidal data and benchmarks; Present-day cliff retreat rate; Present-day down- wearing rate; Measurements to use for misfit calculation;  Density of surface cover (optional) | Total model time;  Past rate of cliff retreat (multiplier relative to present);  Past rate of down- wearing (multiplier relative to present) (optional);  Surface cover depth (optional) | Total exposure (model time); past rate of cliff retreat relative to present; past rate of down-wearing relative to present; platform profile and cliff position prior to down-wearing and cliff retreat; nuclide concentrations across platform; exposure time across platform; platform submergence time; cumulative rate of down-wearing across profile; cumulative water shielding and topographic shielding; optionally, mean surface cover depth and cumulative cover shielding |
|--------------------------------|--------------------------------------------------------------------------------------------------------------------------------------------------------------------------------------------------------------------------------------------------------------------------------------------------------------------|------------------------------------------------------------------------------------------------------------------------------------------------------------------------------------------|--------------------------------------------------------------------------------------------------------------------------------------------------------------------------------------------------------------------------------------------------------------------------------------------------------------------------------------------------------------------------------------------------------------------------------------------------------------------------------------|
|--------------------------------|--------------------------------------------------------------------------------------------------------------------------------------------------------------------------------------------------------------------------------------------------------------------------------------------------------------------|------------------------------------------------------------------------------------------------------------------------------------------------------------------------------------------|--------------------------------------------------------------------------------------------------------------------------------------------------------------------------------------------------------------------------------------------------------------------------------------------------------------------------------------------------------------------------------------------------------------------------------------------------------------------------------------|

#### 60 3 Data requirements

A model is only as good as the input data. For these models, measured nuclide concentrations, topographic shielding and platform elevations, as well as local sea level history and tidal information, are required.

# 3.1 Cosmogenic nuclide samples

As the model is design to find the best-fit coastal erosion scenario for a given cosmogenic nuclide dataset, details of each measured cosmogenic nuclide sample from a platform transect is required as an input. The following information is required: sample name; latitude (decimal degrees); longitude (decimal degrees); elevation (m above sea level); atmospheric pressure (hPa; if known); distance along transect from cliff (m); sample thickness (cm); bulk density (g cm<sup>-3</sup>); topographic shielding factor (unitless, between 0 and 1); <sup>10</sup>Be concentration (atoms g<sup>-1</sup>; mean and 1 sigma uncertainty); year sample was collected.

Atmospheric pressure at each sample location (latitude, longitude) is derived from the ERA-40 atmospheric model (Uppala et al., 2005), with an elevation-pressure relationship (Radok et al., 1996) instead used if the sample is from <-60 °S (Balco et al., 2008; Stone, 2000).

Additionally, if available, the nuclide concentration of an inheritance sample (atoms g<sup>-1</sup>; mean and 1 sigma uncertainty) should be provided. This cosmogenic inheritance sample should be collected in the near-shore, ideally at the base of a cliff or cave within the cliff, to quantify how much inherited <sup>10</sup>Be is in the rock prior to platform exposure (Hurst et al., 2016).

# 75 3.2 Topographic shielding across the platform

At any point along a platform, a rock surface can be shielded from cosmic rays by the topographic relief of the surrounding terrain. A shielding factor can be calculated to account for this impact on cosmogenic production, with values ranging from 0 (completely shielded by topography) and 1 (production is unaffected by topography).

The user is required to provide these measurements as the distance from cliff (m), and corresponding topographic shielding factor (unitless, 0-1). These values can be obtained from measurements (Dunne et al., 1999) at each rock sample collected or at a series of points across the platform, which are then linearly interpolated across the platform. Alternatively, shielding can be calculated from a Digital Elevation Model (Mudd et al., 2016).

Shielding factors can be calculated using the online calculator described by Balco et al. (2008) (http://stoneage.ice-d.org/math/v3/skyline\_in.html), or by using the iceTEA (Jones et al., 2019) MATLAB© tool Topographic\_shielding (https://github.com/iceTEA-code/Tools/blob/master/Topographic shielding.m).

# 3.3 Elevation profile of the platform

To model and visualise nuclide concentrations across the platform, a two-dimensional elevation profile of the platform from the present-day cliff to the seaward edge is required as an input. This should be provided as the distance from cliff (m) and corresponding elevation (m AOD).

Elevations can be measured points, interpolated points or idealised values. Nuclide concentrations are predicted in the model at the spatial resolution of the profile provided here. While the predicted concentrations are linearly interpolated in the model for visualisation and to assess model fit, ideally the elevation should be measured as accurately as possible at or close to each measured sample.

#### 3.4 Relative sea-level history

As seawater can partially or fully shield the platform from cosmic rays, a sea-level history is used in the model to account for the effects on nuclide production. It is particularly important for slowly-eroding coastlines, where the effects from sea-level change could dominate that from cliff retreat and down-wearing. Sea level would have varied spatially as well as temporally, and therefore a local (or at least regional) relative sea-level history is required; a global mean sea-level history would likely not provide an accurate estimate of platform shielding. This history can be sourced from a proxy-based reconstruction or glacial isostatic adjustment model.

The sea-level history should be provided as time (in years before present, ideally with 'present' being the year that the samples were collected), and corresponding sea level (mean; m relative to present AOD). The model linearly interpolates between data points to derive a continuous sea-level history, with water shielding calculated at each model time interval (section 4.1.4).

#### 3.5 Tidal data and benchmarks

- On top of relative sea level, tides determine how much time a platform surface spends submerged by seawater, and how much water shields that surface. The tidal regime therefore determines nuclide production and resulting nuclide concentrations across a platform (Hurst et al., 2017; Regard et al., 2012). This model uses local tidal data to calculate water shielding through time and space (section 4.1.4). Tidal data is required as the tidal frequency, tidal duration (%), and corresponding elevation (central value of 10-cm vertical bins; m AOD).
- Additionally, a series of tidal benchmarks need to be provided: highest astronomical tide (HAT), mean high water level of spring tides (MHWS), mean high water level of neap tides (MHWN), mean low water level of spring tides (MLWS), mean low water level of neap tides (MLWN). These are used for visualisation and model calculations (see sections 4 and 5).

#### 4. Numerical implementation

115

120

## 4.1 Calculating cosmogenic nuclide concentrations

# 4.1.1 Nuclide concentration in a rock surface

At the present time, the concentration  $(N_{x,k})$  (atoms  $g^{-1}$ ) of nuclide k at point x along the rock platform (equivalent to a surface sample) is given by Eq. (1):

$$N_{x,k} = \frac{P_k}{\lambda_k + \frac{\rho \cdot \varepsilon}{\Lambda}} \cdot \left( 1 - exp \left[ -\left( \lambda_k + \frac{\rho \cdot \varepsilon}{\Lambda} \right) \right] \cdot t_{expo} \right) \tag{1}$$

where  $P_k$  is the nuclide's production rate (atoms g<sup>-1</sup> yr<sup>-1</sup>),  $\lambda_k$  is the nuclide's decay constant (1/years),  $\rho$  is rock density (g cm<sup>-3</sup>),  $\varepsilon$  is the surface erosion rate (cm yr<sup>-1</sup>),  $\Lambda$  is the attenuation length (g cm<sup>-2</sup>), and  $t_{expo}$  is the exposure time (year).

For  $^{10}$ Be, a global production rate calibration dataset (Borchers et al., 2016) scaled to the site location (section 4.1.2) and decay constant of  $4.99 \times 10^{-7}$  corresponding to a half-life of  $1.387 \pm 0.012$  Ma (Chmeleff et al., 2010; Korschinek et al., 2010) is used, with the atmospheric attenuation length calculated dependent on the location of the sample (Sato et al., 2008) and adjusted for lithospheric attenuation (see Marrero et al., 2016).

# 125 **4.1.2 Nuclide production**

The total nuclide production is calculated from the combination of spallogenic, which dominates near the surface, and muonogenic production, which dominates at depth:

$$P_k(t) = S_{EL,\zeta}(p, R_c, t) \cdot S_T \cdot S_W \cdot S_C \cdot P_{ref,s,\zeta,k} \cdot exp\left(\frac{-z}{\Lambda_s}\right) + S_T \cdot S_W \cdot S_C \cdot P_\mu(p, R_c, z)$$
 (2)

where  $S_{EL,\zeta}$  is the time-dependent elevation-latitude scaling factor for a particular scaling model ( $\zeta$ ),  $S_T$  is the shielding factor from topography of the surrounding terrain (see section 4.1.34.1.3 Topographic shielding),  $S_w$  is the shielding factor from water (section 4.1.4),  $S_C$  is the shielding factor from surface cover (section 4.1.5),  $P_{ref,s,\zeta}$  is the reference spallogenic (s) production rate (atom  $g^{-1}$  yr<sup>-1</sup>) at present-day sea-level high-latitude (where atmospheric pressure, p = 1013.25) for nuclide k,  $\Lambda_s$  is the effective attenuation length (g cm<sup>-2</sup>), z is the depth (g cm<sup>-2</sup>), and  $P_\mu$  is the production rate (atom  $g^{-1}$  a<sup>-1</sup>) at z due to muons ( $\mu$ ), which is a function of pressure, depth and the cutoff rigidity ( $R_c$ ).

- Three principal scaling models for production by spallation can be used with this model: 1) 'Lm', the time-dependent version of Lal (1991), which uses variations in the dipole magnetic field intensity (Nishiizumi et al., 1989); 2) 'LSD', the time-dependent model of Lifton et al. (2014), which includes dipole and non-dipole magnetic field fluctuations and solar modulation; and 3) 'LSDn', a version of LSD that implements nuclide-specific scaling by incorporating cross-sections for the different reactions (Lifton et al., 2014).
- The geomagnetic history used in all of the time-dependent scaling models includes the CALS3k model for 0-3 ka (Korte et al., 2009; Korte and Constable, 2011), the CALS7k model for 3-7 ka (Korte and Constable, 2005), the GLOPIS-75 model for 7-18 ka (Laj et al., 2004), and the PADM2M model for 18-2000 ka (Ziegler et al., 2011).

For production by muons, the muon flux is scaled using the energy-dependent model of Lifton et al. (2014), within the CRONUScale numerical framework (see Marrero et al., 2016).

#### 145 4.1.3 Topographic shielding

As the production rate is dependent on any shielding of the rock surface (Dunne et al., 1999; Gosse and Phillips, 2001), a time-dependent topographic shielding factor ( $S_T$ ) is used with values ranging from 0 and 1. A typical present-day coastal platform would typically have relatively low values close to a cliff, increasing towards the sea.

In the model, topographic shielding factors are taken from the input data (see sections 3.1 and 3.2) for cases where the cliff position is the same as today. However, where cliff retreat occurs, more seaward points on the platform would have had higher shielding in the past. In this case, topographic shielding is calculated through time as a function of cliff retreat following Swirad et al. (2020):

$$S_T(x) = \frac{\sum_{0}^{t_{max}} f(S_{T,pres,x}, X_{pres,x}, X_{cliff,t}, t_{expo,x})}{t_{expo,x}}$$
(3)

where  $S_{T,pres}$  is the present-day topographic shielding factor at point x along the rock platform,  $X_{pres}$  is the distance along the platform from the present-day cliff position (m),  $X_{cliff}$  is the cliff position at time t in the model run (m), and  $t_{expo}$  is the time of exposure at point x (years ago), with the resulting shielding factor integrated between present day and maximum modelled time ( $t_{max}$ ). In effect, the shielding factor is calculated from the present-day shielding based on the relative distance to the

cliff at each model time, assuming that the cliff retains a constant morphology. Shielding is only calculated for times after the cliff has retreated past each respective point along the platform  $(X_x > X_{cliff})$ .

#### 4.1.4 Water shielding

160

Cosmic rays attenuate through water, reducing nuclide production in underlaying rock. This shielding by seawater therefore needs to be calculated through time and across a platform for accurate nuclide concentration estimates. A water-shielding model is used here (after Swirad et al., 2020), which combines the effects of tidal regime and relative sea-level (RSL) change.

A time-averaged water shielding factor ( $S_W$ , value of 0-1) is calculated at each point along the platform (x):

$$S_W(\mathbf{x}) = \frac{\sum_0^{t_{max}} exp\left(-\frac{z_W(Y_{pres,x},h_W,RSL_t) \cdot \rho_W}{\Lambda}\right)}{t_{expo,i}}$$
(4)

where  $z_W$  is the water depth (cm), which is a function of the present-day elevation of the platform ( $Y_{pres}$ ) at point x, the tidal height ( $h_W$ ), and the relative sea level at time t (years ago), with  $\rho_W$  as the average density of seawater (1.024 g cm<sup>-3</sup>) and  $\Lambda$  as the attenuation length (160 g cm<sup>-2</sup>). For each point along the platform (x), tidal height is determined from the given tidal elevations and durations (see section 3.5), with platform elevations above HAT + RSL assumed to have no shielding from water. As with topographic shielding, water shielding is only calculated for times after the cliff has retreated past each respective point along the platform.

#### 4.1.5 Surface cover shielding

Much like water, any surface material such as beach sand, talus or soil that is covering the platform partially shields the rock from cosmic rays. This can limit nuclide production, reducing <sup>10</sup>Be concentrations in upper portions of platforms. For platforms with fast-retreating cliffs, the reduction in concentration is likely minimal. However, the effect is more significant on slowly-eroding coastlines where high beach widths (>50 m) can absorb wave energy and protect the cliff (Hurst et al., 2017), and likely during periods of prolonged lower relative sea level when the platform could be more of a depositional than erosional environment.

To account for surface cover, a time-averaged shielding factor ( $S_C$ , value of 0-1) is calculated at each point along the platform (x) based on code from iceTEA (Jones et al., 2019):

$$S_C(\mathbf{x}) = \frac{\sum_0^{t_{max}} exp\left(-\frac{z_{C(h_C, X_{pres, x}, Y_{pres, x}, RSL_t)} \cdot \rho_C}{\Lambda}\right)}{t_{expo}i}$$
 (5)

where  $\rho_C$  is the average density of cover (g cm<sup>-3</sup>) and  $z_C$  is the average depth of that cover (cm), which is a function of the given cover thickness ( $h_C$ ), present-day distance along the platform ( $X_{pres}$ ) and elevation of the platform ( $Y_{pres}$ ) at point x, and the relative sea level at time t (years ago). As the existence of surface cover is dependent on relative sea level and the tide,

 $h_C$  is applied to platform elevations above HAT + RSL and which linearly reduces to zero cover thickness at MHWN + RSL, producing a surface cover profile similar to other models (Hurst et al., 2017). The density ( $\rho_C$ ) should be specified, with sand being 1.6 g cm<sup>-3</sup> and soil approx. 1.3 g cm<sup>-3</sup>. As with topographic shielding, cover shielding is only calculated for times after the cliff has retreated past each respective point along the platform.

While more complex mass-shielding approaches more accurately account for production from thermal neutron capture (Delunel et al., 2014; Dunai et al., 2014; Zweck et al., 2013) and variations in cover density with depth (Jonas et al., 2009), a simpler approach is preferred here for computational efficiency, and because spatial and temporal variations in cover depth are unknown.

#### 4.2 Calculating erosion of the platform

#### 4.2.1 Down-wearing

Nuclides produced in the platform surface can be lost from down-wearing (i.e. vertical erosion). This model calculates nuclide concentrations resulting from a constant, increasing or decreasing rate of down-wearing through time, similar to other models (e.g. Hurst et al., 2017; Swirad et al., 2020). However, unlike other models (e.g. Hurst et al., 2017; Regard et al., 2012), down-wearing is not coupled to the rate cliff retreat, ignoring a possible relationship between the two processes, but enabling down-wearing to be explored in the absence of cliff retreat. The rate of surface erosion (down-wearing) over time is calculated from Eq. (6):

$$z_{down}(t) = \rho \cdot \left( \varepsilon_{pres} + \frac{t}{t_{max} - 1} \cdot \left( \begin{cases} \varepsilon_{pres} \cdot |\theta_{down}|, & if \ \theta_{down} < 0 \\ \varepsilon_{pres} \cdot \left( 1 + (\theta_{down} - 1) \right), & if \ \theta_{down} > 0 \\ 0, & if \ \theta_{down} = 0 \end{cases} - \varepsilon_{pres} \right) \right)$$
 (6)

where  $z_{down}(t)$  is erosion in the form of mass depth removed (g cm<sup>-2</sup> yr<sup>-1</sup>),  $\rho$  is rock density (g cm<sup>-3</sup>),  $\varepsilon_{pres}$  is the present-day erosion rate (cm yr<sup>-1</sup>), and  $\theta_{down}$  is an erosion multiplier parameter determining the amount of down-wearing in the past relative to present, which is applied linearly between the maximum modelled time ( $t_{max}$ ) and present day.

Down-wearing is only applied for times after the cliff has retreated past each respective point along the platform, when and where the platform has no surface cover (e.g. beach), and when and where the platform is above MLWN + RSL (i.e. not when a platform surface is submerged by water throughout a year). Episodic, instantaneous jumps in vertical erosion of the platform (i.e. removal of blocks) is not included due to the complex and somewhat random processes involved; however, such an event could be inferred from a cosmogenic nuclide dataset where an outlier exists below an across-platform trend in concentrations, due to the increased loss of nuclides, or in cases of stepped platforms from back-wearing, a series of outliers above and below an across-platform trend (Hurst et al., 2017).

#### 4.2.2 Cliff retreat

Landward retreat of the sea cliff provides the first order control on platform development and, therefore, the timing of platform exposure. Once the cliff has retreated to expose the platform to cosmic rays, it is assumed that nuclide production then initiates in that surface (after accounting for potential cosmogenic inheritance (see Section 3.1; Hurst et al., 2017; Regard et al., 2012; Swirad et al., 2020).

Calculation of cliff retreat in the model follows the approach of Swirad et al. (2020), where each new cliff position ( $X_{cliff}$ ) is based on the previous position and a retreat rate, which can be a constant, increasing or decreasing rate through time (similar to down-wearing; section 4.2.1):

$$220 \quad X_{cliff}(t) = t \cdot \beta_{pres} + \frac{t(t+1)}{2(t_{max}-1)} \cdot \left\{ \begin{cases} \beta_{pres} \cdot \left| \theta_{cliff} \right|, & \text{if } \theta_{cliff} < 0 \\ \beta_{pres} \cdot \left( 1 + (\theta_{cliff} - 1) \right), & \text{if } \theta_{cliff} > 0 \\ 0, & \text{if } \theta_{cliff} = 0 \end{cases} - \beta_{pres} \right\}$$
(7)

where  $\beta_{pres}$  is the present-day cliff retreat rate (m yr<sup>-1</sup>), and  $\theta_{cliff}$  is a retreat multiplier parameter determining the rate of retreat in the past relative to present, which is applied linearly between the maximum modelled time ( $t_{max}$ ) and present day. The cliff retreat rate is assumed to be zero when the lowest point of the platform is above HAT + RSL (i.e. no retreat occurs when the platform is fully elevated above the water throughout a year).

The exposure time  $(t_{expo})$  at each point along the platform (x) is then calculated as:

$$t_{expo}(x) = \begin{cases} t_{max}, & \text{if } X_{cliff}(t_{max}) < X_x \\ t^*, & \text{if } X_{cliff}(t^*) > X_x \end{cases}$$
 (8)

where  $X_x$  is distance along the platform of point x, and  $t^*$  is the first time when a retreating cliff passes that position, defined as:

$$t^* = \min\{t \mid X_{cliff}(t) > X_x\}. \tag{9}$$

## 230 4.3 Optimisation framework to determine best-fit scenario

235

Optimisation is designed to efficiently find a solution, and therefore well-suited to testing hypotheses. Optimisation has proven effective for cosmogenic nuclide applications (Schaefer et al., 2016) as nuclide concentrations in a rock surface can be accurately solved (see section 4.1), allowing for lesser-known environmental variables (e.g. down-wearing and cliff retreat rates) to be numerically explored. An optimisation solver can therefore be used to predict nuclide concentrations from a given model and variables, assess those concentrations against known observations (i.e. measured samples), and then iteratively change the unknown variables to improve the fit with observations. This optimisation can be defined as:

240

$$\Phi_{opt} = \arg \frac{\min}{\Phi} \chi_R^2(\Phi)$$
 (10)

where  $\Phi$  is the set of free parameters (unconstrained variables) being optimised (see Table 1), and  $\chi_R^2$  is the misfit term (see section 4.3.2). The final minimised misfit (i.e. best-fit result) has the smallest misfit between model-predicted and measured nuclide concentrations. Similar to Schaefer et al. (2016), this model uses MATLAB's © fiminsearch optimisation function, which performs multidimensional unconstrained nonlinear minimization using the Nelder-Mead simplex method (Lagarias et al., 1998); note, this requires installation of the Optimization Toolbox.

Optimisation solvers find a local minimum (or *optimum*), which is the smallest misfit value compared to nearby possibilities. However, this result may not be the *global* minimum, which is the smallest misfit value compared to all feasible possibilities.

Therefore, it is advised to use all sub-models and model options (e.g. with surface cover and no surface cover) to explore different possible minimums for the data. Additionally, if the conclusions are based on relatively small differences between scenarios (e.g. rates of past down-wearing or cliff retreat), the sensitively to the choice of optimisation initial values should be explored; an effective approach would be to run the sub-model within a Monte Carlo simulation, evaluating the probability distribution of best-fit outputs from an ensemble of randomised initial values.

#### 250 4.3.1 Cosmogenic inheritance

For the cosmogenic inheritance model (see Table 1), calculation of nuclide production (section 4.1) are used with the assumption of steady-state erosion (denudation) of the cliff-top surface (Hurst et al., 2016). The model-predicted concentration  $N_{k,inh}$  (atoms  $g^{-1}$ ) of nuclide k at depth of the inheritance sample  $(z_{inh})$  is given by the integral:

$$N_{k,inh}(z_{inh}) = \frac{P_{k,z_{inh}}}{\lambda_k + (\frac{\mathcal{E} \cdot \rho}{\rho / \Lambda})}$$
(11)

where  $P_k$  is the combined spallogenic and muonogenic production rate (atoms g<sup>-1</sup> yr<sup>-1</sup>) of the inheritance sample,  $\lambda_k$  is the nuclide's decay constant,  $\rho$  is rock density (g cm<sup>-3</sup>),  $\varepsilon$  is the erosion rate of the cliff-top surface (cm yr<sup>-1</sup>),  $\Lambda$  is the attenuation length (g cm<sup>-2</sup>).

## 4.3.2 Platform erosion

For the platform erosion models, calculations of nuclide production (section 4.1) and platform erosion (section 4.2) are combined to predict nuclide concentrations. The model-predicted average concentration  $N_k$  (atoms  $g^{-1}$ ) of nuclide k in the platform surface (x) at the present time is given by the integral:

$$N_k(\mathbf{x}) = \frac{1}{(z_{x,bottom} - z_{x,top})} \int_0^{t_{expo}(x)} \int_{z_{x,top}}^{z_{x,bottom}} P_k(z + z_{down}(x) \cdot t, t, S_T, S_W, S_C) \cdot \exp(-\lambda_k t) \, dz \, dt \tag{12}$$

265

275

280

where  $z_{x,top}$  and  $z_{x,bottom}$  are the top and bottom mass depths (g cm<sup>-2</sup>) below the platform surface,  $P_k$  is the combined spallogenic and muonogenic production rate (atoms g<sup>-1</sup> yr<sup>-1</sup>) as function of mass depth (z), time (t) and shielding factors  $S_T$ ,  $S_W$  and  $S_C$ ,  $z_{down}$  is the down-wearing rate (g cm<sup>-2</sup> yr<sup>-1</sup>), and  $\lambda_k$  is the decay constant (yr<sup>-1</sup>) of nuclide k.  $N_k$  is zero when the exposure history starts, increasing towards present, with t zero at the present time and positive for past times; the duration of this exposure history is defined by  $t_{expo}$ , which is equivalent to the maximum model time ( $t_{max}$ ) for cases with no cliff retreat.

## 4.3.3 Assessing misfit

The model uses a least-squares misfit statistic to compare nuclide concentrations predicted by the model to measured concentrations, using the measurement uncertainty as the weighting. It is computed as the mean squared residuals, equivalent to a reduced chi-squared ( $\chi_R^2$ ):

$$\chi_R^2 = \frac{1}{n} \sum_{m=1}^n \left( \frac{N_{k,pred}(x_m) - N_{k,meas,m}}{\sigma_{meas,m}} \right)^2 \tag{13}$$

where n is the total number of samples,  $N_{k,pred}$  and  $N_{k,meas}$  are the predicted and mean measured nuclide concentrations for sample m, and  $\sigma_{meas}$  is the corresponding nuclide measurement uncertainty.  $N_{k,pred}$  is taken from the point along the platform (x) where the sample was collected.  $N_{k,meas}$  uses the raw measured concentration minus the concentration of the inheritance sample (section 3.1).

The optimisation solver requires a single misfit value, yet multiple samples are typically measured across a platform. The platform models (zero erosion, down-wearing, cliff retreat and down-wearing; Table 1) therefore have options for how to combine misfit values calculated for each measured sample:

- All samples, which takes a mean of misfits from all measured samples.
  - Minimum only, which uses the minimum measured mean nuclide concentration.
  - Maximum only, which uses the maximum measured mean nuclide concentration.
  - Minimum and maximum, which takes a mean of the misfits from the minimum and maximum measured concentrations, intended for situations where the endmembers are more important than other samples.
- The user should choose the most appropriate misfit method for the hypothesis and sample data. The best-fit result is reported as the  $\chi_R^2$  with degrees of freedom (DOF = n 1), and it is the user's choice about what value is considered acceptable. The critical value can be taken from a chi-squared distribution table using the right-sided tail probability (equal to the significance level, e.g., 0.05 for 95% confidence) and the DOF; for example, the critical value is approximately 12.592 for 6 DOF at 95% confidence.

295

300

## 290 5. Demonstration of model capability

To demonstrate the capability of CCMv1 using a real-world example, each of the CCM models (Table 1) are applied to a published cosmogenic nuclide dataset from a shore platform in North Yorkshire, UK (Swirad et al., 2020). The dataset is well-suited for this purpose as: 1) it shows a common distribution of nuclide concentrations across the platform, with lowest concentrations near the present-day cliff and highest concentrations near the seaward edge, providing a verification of the calculations for nuclide production, water shielding, down-wearing, cliff retreat and beach cover that collectively determine concentrations across the platform; 2) it has some scatter in the concentrations towards the seaward edge, providing a non-idealised test of the optimisation and misfit approach; 3) it has been previously modelled to estimate a cliff retreat rate (Swirad et al., 2020), and CCMv1 has been built on several aspects of that model (topographic shielding, water shielding, down-wearing and cliff retreat), providing an evaluation of model performance. The measured nuclide concentrations, topographic shielding, platform elevation profile, relative sea-level history, tidal data and tidal benchmarks were taken as reported in the original study (see Swirad et al., 2020 for details).

First, the 'cosmogenic inheritance' model is applied to the site's inheritance sample ( $1304 \pm 268$  atoms g<sup>-1</sup>). The model supports the reliability of this sample, showing that the nuclide concentration can be explained by steady-state cliff-top surface erosion rate of <0.001 mm yr<sup>-1</sup> (Fig. 1).

Next, the platform history models are applied to the transect of inheritance-corrected nuclide concentrations. There are several key parameters within these models that impact how nuclide concentrations evolve through time, resulting in differing concentration distributions across a platform that would be expected if measured in surface samples collected today. The relative influence of exposure time, surface cover, down-wearing and cliff retreat is demonstrated in Figure 2. A platform surface that has been exposed to cosmic radiation for more time will have higher concentrations, and the relative difference between the present cliff and seaward edge will increase with time due greater water shielding, dependent on the relative sealevel history. If the platform surface is covered by any material, the nuclide concentrations will be lower; the thicker the cover and/or denser the cover material, the lower the concentrations. This relative reduction in nuclides is largest near the cliff as this part of the platform has experienced more time above HAT, where the model applies thickest cover, and least near the seaward edge of the platform as this part has experienced more time below MHWN, where the model applies no surface cover.

Figure 1. Application of the 'cosmogenic inheritance' model. The model's predicted nuclide concentration is in green, shown with depth for different production pathways (lines) and for total production at the sample depth (circle). The measured inheritance sample has a  $^{10}$ Be concentration of  $1304 \pm 268$  atoms  $g^{-1}$  (mean and standard deviation), which was collected 60 m below the surface of the cliff top. The model perfectly fits the measured concentration (chi-squared = 0) from a best-fit steady-state surface erosion rate of <0.001 (non-zero) mm yr<sup>-1</sup>.

Erosion of the platform also lowers nuclide concentrations—the greater the rate of platform erosion, the lower the concentrations—but the effect varies spatially depending on whether down-wearing or cliff retreat is dominant. Down-wearing produces a relatively consistent nuclide reduction across much of the platform, except near the seaward edge (below MLWN), where less down-wearing occurs in the model over time. Cliff retreat generally produces an increasing concentration trend across the platform, with near-zero concentrations at the present-day cliff and greatest concentrations at the seaward edge, which has a smaller difference between cliff and seaward edge concentrations (i.e. flatter distribution) with increasingly faster retreat rates. In cases where the retreat rate and model time determines the starting cliff position part-way across the platform

(e.g. scenario with a rate of 0.027 m yr<sup>-1</sup> in Fig. 2), concentrations increase in the model from the present-day cliff and the starting position, and then decrease to the seaward edge identical to a no retreat scenario. However, in reality, the seaward part of the platform in that situation would be subject to a longer exposure history and producing an effective jump in concentrations relative to the landward part of the platform (Choi et al., 2012; Regard et al., 2012). This highlights the need to use a long enough relative sea level history when running the model to adequately account for such scenarios of cliff retreat.

The various impacts on nuclide concentrations demonstrated by this model are consistent with that shown in previous studies (e.g. Hurst et al., 2017; Regard et al., 2012). In the case of cliff retreat, nuclide concentrations across a platform can form a characteristic "hump", where midpoints of the platform have higher concentrations than near the cliff or seaward edge because of an optimal balance between exposure duration versus shielding effects (from cliff topography and water) and down-wearing (Hurst et al., 2017). However, the existence, magnitude and location of this hump is dependent on the combination of the various parameters, as well as the site-specific elevation profile of the platform, relative sea-level history and tidal range (Hurst et al., 2017; Regard et al., 2012; Swirad et al., 2020).

The purpose of the CCM models is to determine the best-fit combination of parameters, and now each model will be applied in turn. During each optimised model run, figures are generated for the predicted variables (Fig. 3) and corresponding across-platform predicted nuclide concentrations (Figs. 4-6).

The first to be used is the 'zero platform erosion' model, which can be considered as a test of a null hypothesis: the measured concentrations do not record any erosion, reflecting only relative sea-level change. As the site is located on an erosional coastline, the alternative hypothesis is that the trend of concentrations across the platform records a history of coastal erosion. A decision on how well the best-fit model prediction supports the hypothesis should be made by the user based at least partly on the reduced chi-squared value for a suitable misfit calculation method. In this case, while it is possible to perfectly predict (reduced chi-squared = 0) the minimum or maximum measured concentrations, the fit to the full dataset (min and max range, or mean) is much poorer (reduced chi-squared = 5.57 and 3.51) and is unable to replicate the trend in measured concentrations across the platform (Fig. 4). Therefore, the model's assumption of zero erosion (null hypothesis) can be rejected, supporting the alternative hypothesis that erosion is needed to explain the data.

Figure 2. Impact of different parameters on nuclide concentrations across a real-world platform. The top panel shows shorter to longer exposure time (which corresponds to total model time): 1, 3, 5 and 7 ka. For the rest of the panels, exposure time is 7 ka and only the featured parameter is included and changed. The second panel shows no beach cover and increasing cover depth (density = 1.6 g cm<sup>-3</sup>): 0, 0.5, 1 and 5 m. The third panel shows no down-wearing to increasing down-wearing rates: 0, 0.05, 0.22 (present-day observed rate at the site) and 0.5 mm yr<sup>-1</sup>. The bottom panel shows no cliff retreat to increasing retreat rates: 0, 0.027 (present-day observed rate), 0.05 and 0.5 m yr<sup>-1</sup>. The black symbols are measured concentrations from the site in North Yorkshire UK, and the background shading shows the elevation profile of the platform (Swirad et al., 2020). The same relative sea-level history is used for all scenarios.

**Figure 3. Example of variables used and calculated to predict nuclide concentrations.** This figure is generated by the 'zero platform erosion', 'down-wearing only' and 'cliff retreat and down-wearing' models, replotted for each optimisation iteration, with the final iteration showing the best-fit predictions. The exact variables plotted depend on the model and model options. In this example, the variables correspond to the best-fit prediction from application of the 'cliff retreat and down-wearing' model (see Fig. 6).

Figure 4. Application of the 'zero platform erosion' model. Top panel shows the platform's present elevation profile with tidal benchmarks, and the bottom panel shows the nuclide concentrations (measured and predicted) across that platform. Measured concentrations are plotted as the mean and standard deviation (black symbols), and the model predictions are shown for all points across the platform (green lines). Tidal benchmarks: highest astronomical tide (HAT), mean high water level of spring tides (MHWS), mean high water level of neap tides (MHWN), mean low water level of spring tides (MLWS), mean low water level of neap tides (MLWN). The model was run using an initial value for total model time of 7000 years, based on an assumed exposure history from the relative sea level record. Best-fit predictions are shown for model runs using a misfit against the minimum, maximum, minimum and maximum, and mean of all measured concentrations; the respective results were total model time of 495, 5117, 1612 and 2114 years with reduced chi-squared of 0, 0, 4.57 and 3.51 (with 18 degrees of freedom (DOF)).

Following the model hierarchy, the dataset is next modelled using the 'down-wearing only' model. By running this model, the assumption is that down-wearing must have occurred at a minimum at this site—if the model suitably predicts the concentrations, the platform is the result of down-wearing and no other process (null hypothesis). As the aim is to predict the trend of measured nuclide concentrations across the platform, and the minimum and maximum concentrations are not located at the start and end of the transect, I have chosen to determine the best-fit scenario using the mean of all sample measurements. When applied to the dataset, the down-wearing model has an improved fit relative to the 'zero platform erosion' model (reduced chi-squared = 2.82 vs 3.51), however the across-platform concentration trend is still not captured (Fig. 5). Accounting for down-wearing produces a best-fit scenario of a longer exposure time with slightly higher concentrations towards the seaward edge of the platform; but it still predicts a broadly constant distribution of concentrations (no trend) across the platform. The model can also test if the concentrations are additionally the result of beach cover (or potentially other types of surface cover). When included, the model predicts a slightly better fit (reduced chi-squared of 2.36), causing the concentrations to be relatively lower near the cliff (above HAT), and relatively higher towards the seaward edge. Neither application of the model (with or without beach cover) captures the measured trend and therefore the null hypothesis is again rejected, implying another process such as cliff retreat must explain the data.

Finally, the 'cliff retreat and down-wearing' model is applied. In combination with the 'down-wearing only' model, it can be used to support an alternative hypothesis that the nuclide concentrations record cliff retreat. Fitted the dataset, the model predicts the measured concentrations much better than the 'zero platform erosion' and 'down-wearing only' models (reduced chi-squared <1), capturing the trend of lower concentrations near the cliff and higher concentrations near the seaward edge of the platform (Fig. 6). The result therefore supports this alternative hypothesis, agreeing with the original study that this shore platform in North Yorkshire was formed principally by cliff retreat during the Holocene (Swirad et al., 2020).

The 'down-wearing only' and 'cliff retreat and down-wearing' models can also be used to test hypotheses related to rates of erosion—for example, the past rate of cliff retreat and/or down-wearing as recorded by these concentrations was no different than at present (null hypothesis). In this North Yorkshire application, the best-fit predicted cliff retreat rate was the same as present (2.7 cm yr<sup>-1</sup>), supporting the null hypothesis and the conclusion of the original study that the cliff has been steadily retreating.

One difference between the CCMv1 result and that of the original study is the exact best-fit cliff retreat rate. The modelling in the original study simulated platform histories for the length of the known relative sea-level history (7000 years), determining a retreat rate of  $4.5 \pm 0.63$  cm yr<sup>-1</sup>. In CCMv1, the model time and starting cliff position are unconstrained, resulting in a best-fit scenario that had the cliff position inland of the seaward edge of the platform, corresponding to a model time slightly less than the original study (6596 years) and the present-day retreat rate. In cases of a sufficiently slow cliff retreat rate, sufficiently long model time, and sufficiently lower-than-present relative sea level in the past, the model could predict a starting cliff position inland of the seaward edge of the platform; this platform geometry would produce a step in nuclide concentrations

between the landward and seaward sides of the starting cliff position. For this dataset, however, the modelled cliff position does not have a substantial impact on the seaward nuclide concentrations, producing a result that is similar, although not identical, to the original study. Nevertheless, to more fully explore the exposure and erosion history and, in turn, better constrain plausible cliff retreat rates, a longer relative sea-level history should be used.

In summary, the application of CCMv1 models to the North Yorkshire cosmogenic nuclide dataset demonstrates an effective hierarchical and hypothesis-testing approach for investigating the history of rocky shore platforms. The models support the reliability of the inheritance sample, and the finding that the platform transect records cliff retreat at a steady rate similar to present over the last 6.5-7 thousand years, highlighting the reliability of the model to predict nuclide concentrations and determine best-fit scenarios.

Figure 5. Application of the 'down-wearing only' model. The model-generated figure is like that of the 'zero platform erosion' model (Fig. 4), but with the predicted platform elevation profile before down-wearing additionally plotted. Here the model was run with and without surface (beach) cover, using an initial value for total model time of 7000 years, a present-day down-wearing rate of 0.22 mm yr<sup>-1</sup> and an initial value for down-wearing change relative to present of 0 (same/constant rate). For the run with surface cover, a density value of 1.6 g cm<sup>-3</sup> (sand) and an initial value for cover depth of 1 m was additionally used. The top panel shows the best-fit platform elevation without cover, and bottom panel with cover. Best-fit predictions are shown in the middle panel for model runs using a misfit against the mean of all measured concentrations, with a total model time and relative down-wearing rate for the run without surface cover of 3460 years and 0.0 times faster than present (reduced chi-squared of 2.82 for 18 DOF), and total model time, relative down-wearing rate and mean cover depth for the run with surface cover of 6389 years, 0.22 mm yr<sup>-1</sup> (same as present) and 5.0 m (reduced chi-squared of 2.36 for 18 DOF). Note, the best-fit scenario with surface cover produces overall relatively higher concentrations because the model time is greater than the best-fit scenario without surface cover.

**Figure 6. Application of the 'cliff retreat and down-wearing' model.** See Fig. 4 and 5 for description of the panels, and note that the solid grey line with cliff in the top panel represents the predicted platform profile at the start of the model. Here the model was run using an initial value for total model time of 7000 years, present-day cliff retreat rate of 0.027 m yr<sup>-1</sup>, present-day down-wearing rate of 0.22 mm yr<sup>-1</sup>, initial values for cliff retreat and down-wearing change relative to present of 0 (same/constant rates), and surface cover density value of 1.6 g cm<sup>-3</sup> and initial depth value of 0.5 m. The best-fit prediction is shown that uses a misfit against the mean of all measured concentrations, with a total model time of 6596 years, cliff retreat rate same as present and beach cover of 0.81 m (reduced chi-squared of 0.41 for 18 DOF).

#### 6 Scope, limitations and scalability

CCM is designed to quantify the formation history of rocky shore platforms by determining plausible scenarios of cliff retreat and down-wearing from measured in situ cosmogenic nuclide concentrations. Similar to other models, it explicitly simulates the combined effects of sea-level change, tidal regime, beach cover, and topographic and water shielding on time-dependent nuclide production. By allowing erosion and exposure histories to vary flexibly in space and time, CCM can represent complex coastal evolution scenarios that may not conform to simplified process-based formulations.

The model suited to various applications. First, it can be used for inverse modelling of platform erosion histories, for example, to estimate long-term erosion rates—how fast did cliff retreat and/or down-wearing occur in the past? Was this faster, slower or similar to today? Second, it can also be used in rocky shore settings where no cliff retreat occurs, which differs from all previous models—for example, in resistant lithologies where the shore profile is relatively steep and convex, or where erosion solely occurs as down-wearing and perhaps driven by episodic (e.g. tectonic) changes in sea level. Third, CCM is particularly useful for hypothesis testing of sea-level, erosion or beach cover scenarios, where the goal is to identify the most likely combination of parameters that describe platform formation. This is most powerful when the hypotheses are informed by other evidence—such evidence could be related to any of the input data or model parameters, for example, testing if the measured concentrations support Holocene cliff retreat, present-day down-wearing rates or tectonically-controlled relative sea-level change. Finally, the model is effective for exploratory simulations that examine the sensitivity of platform nuclide inventories to different rates and combinations of cliff retreat, down-wearing or cover—for example, are the findings still valid if beach cover is considered, or down-wearing without cliff retreat is considered?

There are some limitations to CCMv1, which should be considered for the specific application. First, the model does not explicitly simulate the physical drivers of erosion (e.g. wave energy). In cases where the aim is to investigate the role of specific physical drivers on erosion, particularly where data exist to inform associated parameters, dynamic process-based numerical models are more appropriate (e.g. Hurst et al., 2017). Second, as with all other rocky shore models, CCM evaluates platform

history in a transect, assuming that all processes determining nuclide concentrations operate perpendicular to the coast. It must therefore be applied to study sites with minimal process and geomorphic variability in the along-coast direction.

Finally, while CCMv1 identifies best-fit scenarios within a discrete parameter framework, it is important to recognise that equifinality—where multiple parameter combinations yield similar model outputs—can still occur. This is particularly relevant when datasets are sparse or when erosion histories are complex. CCMv1 helps mitigate equifinality by allowing users to test alternative hypotheses across a suite of models, and by enabling sensitivity analyses through varied misfit methods and optimisation initial values. Users can further explore equifinality by implementing ensemble or Monte Carlo simulations, as outlined in Section 4.3. Future versions of CCM could incorporate multi-objective or Bayesian optimisation approaches (Shadrick et al., 2021) to better constrain parameter uncertainty and improve robustness in cases where equifinality is high.

Building on this, several avenues exist for enhancing CCM to improve its flexibility and applicability across diverse coastal settings. Currently CCMv1 can only be used for <sup>10</sup>Be concentrations, which is most widely used and is well suited to platform erosion, particularly over the Holocene. But the code could be extended for use with other nuclides (e.g. <sup>3</sup>He, <sup>14</sup>C, <sup>21</sup>Ne, <sup>26</sup>Al and <sup>36</sup>Cl). In particular, the short half-life and relatively large surface muonogenic production of <sup>14</sup>C (Hippe, 2017) makes the nuclide well suited to constrain erosion and shielding histories, particularly in combination with <sup>10</sup>Be. A key strength of CCM is that it is built on CRONUScalc, which includes these other nuclides. Similarly, any important revisions of the existing geomagnetic databases, production rates and scaling models could be easily updated in CCM, as these are typically coded in MATLAB©, and it is expected that future updates would be consistent with CRONUScalc and its predecessors. Additionally, the calculation framework used in CCM allows for the calculation of time-dependent nuclide production at depth. CCM could be adapted to account for vertical changes in production and erosion in the case of complex scenarios, as would occur from a multiphase cliff retreat (Regard et al., 2012), and for measured concentrations in depth profiles.

As with other cosmogenic nuclide models (e.g. Hurst et al., 2017; Regard et al., 2012; Swirad et al., 2020), CCMv1 uses mean relative sea level to prescribe the sea-level history. However, this variable has uncertainty, and for some study regions it may be unknown. The magnitude and across-platform distribution of nuclide concentrations differs between stable, rising and falling scenarios (Fig. 7), determined by how much each part of the platform is shielded by water. A future version of CCM could account for the uncertainties in relative sea-level data derived from proxy-based reconstructions or glacial isostatic adjustment models. For fast-eroding sites, this would provide a more rigorous quantification of cliff retreat and down-wearing rates. For slowly-eroding sites, this would provide a test of plausible sea-level histories and enable reconstruction of relative sea-level change. It would be most effective where cosmogenic nuclide samples record a signal of platform emergence (e.g. Bierman et al., 2018), as scenarios of falling sea level are more easily identifiable (Fig. 7).

Figure 7. Impact of relative sea level on a non-eroding rocky shore. Top panel shows seven simplified sea-level histories over the last 7000 years: stable (grey), rising linearly (by 5 m solid light blue, and by 10 m solid dark blue), rising with upward/downward step change (dashed/dotted light blue), falling linearly (by 5 m solid light red, and by 10 m solid dark red), and falling with downward/upward step change (dashed/dotted light red). Bottom panel shows the corresponding nuclide concentrations across the platform. No surface cover, down-wearing or cliff retreat is applied for these scenarios. The overall magnitude and across-platform distribution of nuclide concentrations differs between stable, linearly rising and linearly falling scenarios. Adding a step change to the trend has an impact, albeit relatively small. Differences between scenarios of rising sea level are smaller than between those of falling sea level as the platform is influenced less by water shielding over time.

https://doi.org/10.5194/egusphere-2025-4086 Preprint. Discussion started: 28 October 2025

© Author(s) 2025. CC BY 4.0 License.

#### 7 Conclusion

CCMv1 offers a flexible and modular framework for reconstructing the history of rocky shore platforms using cosmogenic nuclide concentrations. Rather than simulating physical erosion processes directly, CCMv1 focuses on quantifying plausible exposure and erosion scenarios through forward modelling and empirical optimisation. Its suite of sub-models enables structured hypothesis testing, allowing users to explore platform histories ranging from stable, non-eroding surfaces to dynamic cliff retreat. This makes CCMv1 particularly well-suited for investigating the relative influence of different erosion mechanisms, as well as sea-level change, tidal regime, and surface cover on nuclide inventories.

The demonstration of CCMv1 using a published dataset from North Yorkshire, UK, highlights its capability to reproduce measured nuclide concentrations and infer a consistent history of Holocene cliff retreat. The model's integration with CRONUScale ensures compatibility with global production rate calibrations, while its optimisation framework supports sensitivity testing and misfit evaluation across multiple scenarios.

Importantly, CCMv1 is not intended to replace existing process-based coastal evolution models. Instead, it provides an alternative framework for interpreting cosmogenic nuclide data and investigating rocky shore platform histories, particularly in settings where physical drivers are poorly constrained or where erosion may be minimal. The current implementation is well-suited to reconstructing platform exposure histories and testing erosion scenarios across a range of coastal morphologies. Looking ahead, the model hierarchy and nuclide calculation framework of CCMv1 offer a foundation for future developments, including implementations that can simulate more complex (e.g. multi-stage) exposure—erosion histories and test competing relative sea-level scenarios. By enabling structured hypothesis testing and flexible inversion of platform histories, CCMv1 contributes a novel tool to the research community—advancing the capacity to quantify long-term coastal change and supporting efforts to understand rocky coast evolution in the context of environmental change.

# Code availability

The software used to compute the numerical solutions has been written in MATLAB (MathWorks Inc., 2023) and is available at https://doi.org/10.5281/zenodo.15454613.

#### **Author contribution**

RSJ designed the model framework, formulated the numerical implementation, ran analyses using the models to demonstrate capability, produced the figures and wrote the manuscript.

# **Competing interests**

475 The author declares that they have no conflict of interest.

## Acknowledgements

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
