# Peer review of "Coastal-Cosmo-Model (CCMv1): a cosmogenic nuclide model for rocky coastlines"

_EGUsphere, 2025_

## Author Response (AR1)

**Changes to the manuscript in response to reviewer comments**

Please see my reply to each reviewer for my full response. Here I outline the specific changes to the manuscript that address reviewer comments.

**RC1**

- Paragraph added in section 1 (Introduction) to summarise previous contributions to the field:

    *"To date, the number of numerical models that directly link in situ cosmogenic nuclide concentrations to the evolution of rocky coasts remains relatively small. Early and subsequent studies have demonstrated how cliff retreat and platform lowering imprint characteristic across-platform concentration patterns, and have used forward modelling to explore the influence of processes such as retreat rate and down-wearing (e.g. Hurst et al., 2017; Regard et al., 2012). These approaches have been further applied to infer long-term coastal erosion rates by fitting model predictions to measured datasets, particularly for retreat-dominated coastlines (e.g. Hurst et al., 2016; Swirad et al., 2020), with recent developments incorporating more formal optimisation strategies to constrain retreat histories (Shadrick et al., 2021). Collectively, these studies highlight the power of cosmogenic nuclides for quantifying rocky coast evolution, but also point to the need for modelling frameworks that are flexible, modular, and explicitly designed for hypothesis testing across a wider range of coastal morphologies and erosion regimes."*

- Paragraph added in section 2 to clarify and justify down-wearing only model:

    *"Down-wearing is treated here as a more fundamental and ubiquitous erosional process on rocky coasts, as vertical lowering of a platform may occur even where long-term cliff retreat is negligible (e.g. resistant or convex shore platforms). The 'down-wearing only' model therefore serves as a key intermediate hypothesis, allowing the sensitivity of nuclide concentrations to vertical erosion and relative sea-level history to be tested independently of cliff retreat. A standalone 'cliff-retreat-only' model is not implemented because cliff retreat would generally be expected to coincide with some degree of platform lowering. However, users may effectively explore a cliff-retreat-only scenario by setting the down-wearing rate to zero within the combined 'cliff retreat and down-wearing' model."*

- Sentence added in section 6 to highlight potential modifications to allow for more complex applications:

    *"While CCMv1 represents temporal changes in cliff retreat and down-wearing rates using simple linear multipliers relative to present-day values, the underlying model structure readily permits more complex parameterisations, including non-linear or user-defined rate histories, which may be appropriate for specific sites or hypotheses, and represent a natural direction for future development."*

**RC2**

- Subsection added to describe the modular structure of the framework:

  *"4.1 Modularity and extensibility of the modelling framework*

  *CCM was intentionally designed as a modular numerical framework to facilitate modification and extension, while retaining a simple optimisation interface for users. The CCM framework is organised such that model control and configuration, optimisation and model execution, misfit evaluation, and generic nuclide and shielding calculations are handled by distinct components, allowing individual elements (e.g. erosion or shielding formulations) to be modified or replaced independently without altering the optimisation scheme.*

  *The optimisation routine itself places no constraints on the functional form of erosion or shielding equations, provided that model predictions can be generated consistently for a given set of parameters and compared to measured nuclide concentrations via a scalar misfit function. As such, assumptions such as spatially uniform down-wearing rates, linear temporal scaling of erosion rates, or cessation of cliff retreat above a given tidal elevation are modelling choices rather than requirements of the optimisation scheme. Users may modify or replace individual components of the framework—for example by introducing elevation-dependent erosion rates, alternative retreat formulations, or different shielding parameterisations—without altering the optimisation approach, provided the modified model returns predicted nuclide concentrations for a given parameter set.*

  *More substantial changes that require additional inputs or free parameters are best implemented as new model variants built on the existing CCMv1 structure. In this way, CCMv1 is intended to provide a foundation for study-specific and process-motivated extensions rather than a closed or prescriptive model formulation."*

- Terms of HAT, MHWN and MLWN have been redefined in sections 4 and 5 to remind reader of the acronyms.

- Sentence added near the start of section 2 to explain an inheritance sample:

  *"As a cosmogenic inheritance sample is often measured at a study site to quantify how much inherited 10Be is in the rock prior to platform exposure, the reliability of this sample should be assessed."*

- Wording of "past rate" modified in section 2 to avoid confusion, including a footnote added to Table 1:

  *"Past rates are implemented as linear multipliers applied between the maximum modelled time and the present-day rate (see Eqs. 6–7)."*

- The timescale of the modelling exercise has been added to the first paragraph of section 5, referring to the *"~6.5 kyr"* evolution originally modelled by Swirad et al. (2020).

- Sentence added to the Introduction to clarify the process-agnostic design of CCM:

  *"Rocky shore platform evolution is driven by multiple physical processes (e.g. wave erosion and salt weathering), but the framework presented here does not model these drivers explicitly, instead representing their integrated, first-order effect on platform exposure and erosion inferred from cosmogenic nuclide concentrations."*

- CCM is now defined (as *Coastal-Cosmo-Model*) in the Introduction.

- Sentence reworded at the start of section 3:

  *"In data-driven applications, model performance is strongly influenced by the quality of the input data."*

- Typo corrected to "*designed*".

- All references of "from cliff" changed to "*from cliff base*".

- 'AOD' defined as "*above ordnance datum*".

- Sentence added to describe tidal duration (%):

  *"tidal duration is the percentage of the time that the tide is in that elevation bin, with a value provided for each elevation bin."*

- Water shielding equation (Eq. 4) has been re-written to more accurately represent how the tidal parameters are used. For clarity and completeness, it has been split into two equations, accompanied by revised symbol description:

  *"A time-averaged water shielding factor ($S_W$, value of 0-1) is calculated at each point along the platform (x):*

  $$S_W(x) = \frac{1}{t_{expo,i}} \sum_{t=0}^{t_{max}} \left\langle exp\left(-\frac{z_W\,(Y_{pres}(x),h,RSL_t)\,\cdot\,\rho_W}{\Lambda}\right)\right\rangle_{tide}$$
  *(4)*

  *where $z_W$ is the water depth (cm), which is a function of the present-day elevation of the platform ($Y_{pres}$) at point $x$, the tidal elevation (h), and the relative sea level at time t (years ago), with $\rho_W$ as the average density of seawater (1.024 g cm$^{-3}$) and $\Lambda$ as the attenuation length (160 g cm$^{-2}$).*

  *The operator $\langle\cdot\rangle_{tide}$ denotes an average over the tidal cycle, computed from discrete tidal elevation bins and their associated durations:*

  $$\langle f(h)\rangle_{tide} = \sum_{k=1}^{N} p_k\, f(h_k), \qquad p_k = d_k/100$$
  *(5)*

  *where $h_k$ is the central elevation of the k-th 10-cm tidal bin, and $d_k$ is the percentage of time spent in that bin ($\sum_k p_k = 1$).*

  *For each point along the platform (x) and at each model time step, water depth and attenuation are evaluated for each tidal elevation bin and averaged over the tidal cycle using fractional durations (see section 3.5). For the bin containing the platform elevation, the contribution is linearly portioned between dry and submerged fractions across the 10-cm bin width ($\pm5$ cm) to avoid step changes in shielding. Platform elevations above highest astronomical tide (HAT) + RSL are assumed to experience no shielding from water. As with topographic shielding, water shielding is only calculated for times after the cliff has retreated past each respective point along the platform."*

- Typo corrected to "*sensitivity*".

- Figure 2 panels labelled a, b, c and d. Caption text updated accordingly.

- Typo corrected to "*Fitted to the dataset*".

- Typo corrected to "*optimisation of initial values*".

- Paragraph added to section 6 to outline how CCM can be used as a tool to inform sampling strategy:

*"Beyond retrospective analysis of measured datasets, CCM can also be used in a forward, exploratory mode to inform sampling strategies prior to fieldwork. By specifying hypothetical erosion, exposure and shielding scenarios, the framework can be used to predict expected cosmogenic nuclide concentrations across a platform without requiring measured concentrations to constrain an optimisation. Such simulations allow users to explore how different processes (e.g. down-wearing versus water or topographic shielding) influence concentration patterns, assess the sensitivity of nuclide inventories to key parameters, and identify locations along a platform where samples are most likely to be diagnostic. This capability may be particularly useful for evaluating potential equifinality and for designing sampling transects that maximise the ability of cosmogenic nuclide data to discriminate between competing hypotheses of platform evolution."*